# DeepCrawl: Deep Reinforcement Learning for Turn-based Strategy Games

**Alessandro Sestini** [1]   **Alexander Kuhnle** [2]   **Andrew D. Bagdanov** [1]

## Abstract

In this paper we introduce DeepCrawl, a fully-playable Roguelike prototype for iOS and Android in which all agents are controlled by policy networks trained using Deep Reinforcement Learning (DRL). Our aim is to understand whether recent advances in DRL can be used to develop convincing behavioral models for non-player characters in videogames. We begin with an analysis of requirements that such an AI system should satisfy in order to be practically applicable in video game development, and identify the elements of the DRL model used in the DeepCrawl prototype. The successes and limitations of DeepCrawl are documented through a series of playability tests performed on the final game. We believe that the techniques we propose offer insight into innovative new avenues for the development of behaviors for non-player characters in video games, as they offer the potential to overcome critical issues with classical approaches.

## 1. Introduction

In recent decades the videogame industry has seen consistent improvement in the production of quality games and today it competes economically with the most important multimedia industries. Game companies have a major impact on the economy through the sales of systems and software, and in fact the revenue of the videogame industry in 2018 was estimated to be more than *twice* that of the international film and music industries combined. This market is expected to be worth over 90 billion dollars by 2020. Compared to the not so distant past when gaming consoles and PC gaming were not so diffused, videogames

are no longer a niche but are instead transversal across ages, genders and devices. Today there are more than 2.5 billion gamers worldwide, this especially thanks to the advent of mobile games and more accessible consoles (WePC, 2019).

Technological advances in the industry have resulted in the production of increasingly complex and immersive gaming environments. However, the creation of Artificial Intelligence (AI) systems that control non-player characters (NPCs) is still a critical element in the creative process that affects the quality of finished games. This problem is often due to the use of classical AI techniques that result in predictable, static, and not very convincing NPC strategies. Reinforcement learning (RL) can help overcome these issues providing an efficient and practical way to define NPC behaviors, but its real application in production processes has issues that can be orthogonal to those considered to date in the academic field: How can we improve the gaming experience? How can we build credible and enjoyable agents? How can RL improve over classical algorithms for game AI? How can we build an efficient ML model that is also usable on all platforms, including mobile systems?

At the same time, recent advances in Deep Reinforcement Learning (DRL) have shown it is possible to train agents with super-human skills able to solve a variety of environments. However the main objective of DRL of this type is training agents to mimic or surpass human players in competitive play in classical games like Go (Silver et al., 2016) and video games like DOTA 2 (OpenAI, 2019). The resulting, however, agents clearly run the risk of being far too strong, of exhibiting artificial behavior, and in the end not being a *fun* gameplay element in a playable product.

Video games have become an integral part of our entertainment experience, and our goal in this work is to demonstrate that DRL techniques can be used as an effective *game design* tool for learning compelling and convincing NPC behaviors that are natural, though not superhuman, while at the same time provide challenging and enjoyable gameplay experience. As a testbed for this work we developed the DeepCrawl Roguelike prototype, which is a turn-based strategy game in which the player must seek to overcome NPC opponents and NPC agents must learn to prevent the player from succeeding. We emphasize that our goals are different than those of AlphaGo and similar DRL systems

[1]Dipartimento di Ingegneria dell'Informazione, Università degli Studi di Firenze, Florence, Italy [2]Department of Computer Science and Technology, University of Cambridge, United Kingdom. Correspondence to: Alessandro Sestini <alessandrosestini92@gmail.com>, Andrew D. Bagdanov <andrew.bagdanov@unifi.it>.

*Proceedings of the 36th International Conference on Machine Learning*, Long Beach, California, PMLR 97, 2019. Copyright 2019 by the author(s).

applied to gameplay: for us it is essential to limit agents so they are beatable, while at the same time training them to be convincingly competitive. In the end, playing against superhuman opponents is not *fun*, and neither is playing against *trivial* ones. This is the balance we try to strike.

The rest of the paper is organized as follows. In the next section we give a brief overview of techniques applied to game AIs, and the current state-of-the-art of DRL applied to video games. In section 3 we describe the game environment and the gameplay mechanics of DeepCrawl, and also delineate the requirements that an AI system should satisfy in order to be practically applied in videogame production. In section 4 we propose a DRL model used in DeepCrawl, and follow in section 5 with the implementation details of the model. We report on results of a series of playability tests performed on DeepCrawl, and we conclude in section 7 with a discussion of our results and by indications of important open challenges.

## 2. Related work

Game AI has been a critical element in video game production since the dawn of this industry; agents have to be more and more realistic and intelligent to provide the right challenge and level of enjoyment to the user. However, as game environments have grown in complexity over the years, scaling traditional AI solutions like Behavioral Trees (BT) and Finite State Machines (FSM) for such complex contexts is an open problem (Yannakakis & Togelius, 2018).

Reinforcement Learning (RL) (Sutton et al., 1998) is directly concerned with the interaction of agents in an environment. RL methods have been widely used in many disciplines, such as robotics and operational research, and games. The breakthrough of applying DRL by DeepMind in 2015 (Mnih et al., 2015) brought techniques from supervised Deep Learning (such as image classification and Convolutional Neural Networks) to overcome core problems of classical RL. This combination of RL and neural networks has led to successful application in games. In the last few years several researchers have improved upon the results obtained by DeepMind. For instance, OpenAI researchers showed that with an Actor Critic (Konda & Tsitsiklis, 2003) algorithm such as Proximal Policy Optimization (PPO) (Schulman et al., 2017) it is possible to train agents to superhuman levels that can win against professional players in complex and competitive games such as DOTA 2 (OpenAI, 2019) and StarCraft (DeepMind, 2019).

As already discussed in the introduction, most of the works in DRL aim to build agents replacing human players either in old-fashioned games like Go or chess (Silver et al., 2016; Asperti et al., 2018) or in more recent games such as Doom or new mobiles games (OpenAI, 2019; Vinyals et al., 2017;

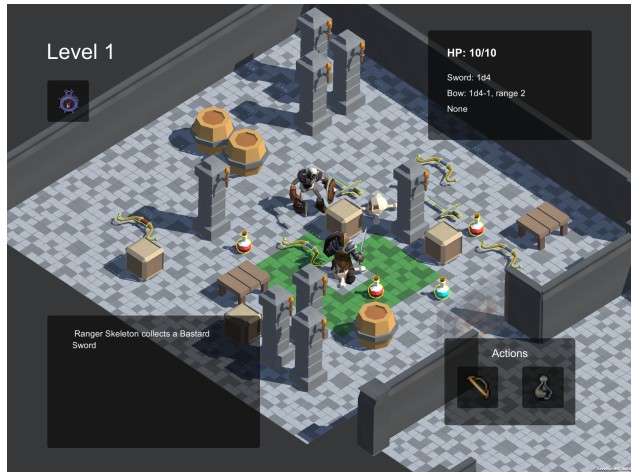

*Figure 1.* Screenshot of the final version of DeepCrawl. Each level of the game consists of one or more rooms, in each of which there is one or more agents that must be defeated. In each room there is collectible loot that can help both the player and the agents. To clear a level the player must fight and win against all the enemies in the dungeon. The game is won if the player completes ten dungeon levels.

Oh et al., 2019; Kempka et al., 2016; Juliani et al., 2019). Our objective, however, is not to create a new AI system with superhuman capabilities, but rather to create ones that constitute an active part of the game design and gameplay experience. In the next section we define the main characteristics of the game created for this purpose, with an overview of the fundamental requirements that a ML system must satisfy to support an enjoyable gaming experience.

## 3. Game design and desiderata

In this section we describe the main gameplay mechanics of DeepCrawl and the requirements that the system should satisfy in order to be used in a playable product. The DeepCrawl prototype is a fully playable Roguelike game and can be downloaded for Android and iOS [1]. In figure 1 we give a screenshot of the final game.

### 3.1. Gameplay mechanics

DeepCrawl is a Roguelike game. The term *Roguelike* refers to a particular type of turn-based roleplaying game, typically 2D with a third-person perspective. The Roguelike genre was born in 1980 with the game *Rogue*, and Roguelikes are experiencing a sort of renaissance in the gaming community. In 2019 we find several hundreds of games claiming to be Roguelikes in the Steam catalog.

---

[1] Android Play: http://tiny.cc/DeepCrawl
App Store: http://tiny.cc/DeepCrawlApp

There are several aspects of Roguelikes that make them an interesting testbed for DRL as a game design tool. A few characteristics of particular note for us are that Roguelikes:

- are generally considered to be a **single-player game**, which reduces the actions and interactions to those between player and environment;

- have a **procedurally created environment** with random elements, which makes them especially suitable to long-term, episodic training;

- are **turn-based** in that time passes only when an actor makes an action, which simplifies the action and environment model;

- are **non-modal** in that every action is available for the actors regardless the level of the game; and

- are focused on a **hack-and-slash** gameplay, focusing on killing opponents encountered during exploration, which lends itself well for defining sparse rewards for training.

In fact, Roguelikes are often used as a testbed game genre specifically because they involve a limited set of game mechanics, which allows game designers to concentrate on emergent complexity of gameplay as a combination of the relatively simple set of possibilities given to the player.

The primary gameplay mechanics in DeepCrawl are defined in terms of several distinct, yet interrelated elements.

**Actors.** Success and failure in DeepCrawl is based on direct competition between the player and one or more agents guided by a deep policy network trained using DRL. Player and agents act in procedurally generated rooms, and player and agents have exactly the same characteristics, can perform the same actions, and have access to the same information about the world around them.

**Environment.** The environment visible at any instant in time is represented by a random grid with maximum size of $10 \times 10$ tiles. Each tile can contain either an agent or player, an impassible object, or collectible loot. Loot can be of three types: melee weapons, ranged weapons, or potions. Moreover, player and agent are aware of a fixed number of personal characteristics such as HP, ATK, DEX, and DEF (used in practically all Roguelike games). Agents and player are also aware of their inventory in which loot found on the map is collected. The inventory can contain at most one object per type at a time, and a new collected item replaces the previous one. The whole dungeon is composed of multiple rooms, where in each of them there are one or more enemies. The range of action of each NPC agent is limited to the room where it spawned, while the player is free to move from room to room.

**Action space.** Each character can perform 17 different discrete actions:

- **8 movement actions** in the horizontal, vertical and diagonal directions; if the movement ends in a tile containing another agent or player, the actor will perform a melee attack: this type of assault deals random damage based on the melee weapon equipped, the ATK of the attacker, and the DEF of the defender;

- **1 use potion action**, which is the only action that does not end the actor's turn. DeepCrawl has two buff potions available, one that increases ATK and DEF for a fixed number of turns, and heal potion that heals a fixed number of HP; and

- **8 ranged attack actions**, one for each possible direction. If there is another actor in selected direction, a ranged attack is performed using the currently equipped ranged weapon. The attack deals a random amount of damage based on the ranged weapon equipped, the DEX of the attacker, and the DEF of the defender.

### 3.2. Desiderata

As defined above, our goals were to create a playable game, and in order to do this the game must be enjoyable from the player's perspective. Therefore, in the design phase of this work it was fundamental to define the requirements that AI systems controlling NPCs should satisfy in order to be generally applicable in videogame design:

**Credibility.** NPCs must be *credible*, that is they should act in ways that are predictable and that can be interpreted as intelligent. The agents must offer the right challenge to the player and should not make counterintuitive moves. The user should not notice that he is playing against an AI.

**Imperfection.** At the same time, the agents must be *imperfect* because a superhuman agent is not suitable in a playable product. In early experiments we realized that it was relatively easy to train *unbeatable* agents that were, frankly, no fun to play against. It is important that the player always have the chance to win the game, and thus agents must be beatable.

**Model-free.** Enemy agents must be *model-free* in that developers do not have to manually specify strategies – neither by hard-coding nor by carefully crafting specific rewards – specific to the game context. The system should extrapolate strategies independently through the trial-and-error mechanism of DRL. Moreover, this model-free system should generalize to other Roguelike games sharing the same general gameplay mechanics.

**Variety.** It is necessary to have a certain level of *variety*

in the gameplay dynamics. Thus, it is necessary to support multiple agents during play, each having different behaviors. The system must provide simple techniques to allow agents to extrapolate different strategies in the training phase.

These basic gameplay mechanics and broad desiderata provide the context of the DeepCrawl design and implementation. In the following we will see our proposed DRL model, and in particular the main elements chosen specifically to satisfy these requirements.

## 4. Proposed model

Here we describe in detail the main elements of the DRL model that controls the agent in DeepCrawl, with particular attention to the neural network architecture and the reward function.

### 4.1. Policy network and state representation

We used a policy-based method to learn the best strategy for agents controlling NPCs. For these methods, the network must approximate the best policy. The neural network architecture we used to model the policy for NPC behavior is shown in figure 2. The network consists of four input branches:

- the first branch takes as input the whole map of size $10 \times 10$, with the discrete map contents encoded as integers:

    - 0 = impassable tile or other agent;
    - 1 = empty tile;
    - 2 = agent;
    - 3 = player; and
    - 4+ = collectible items.

    This input layer is then followed by an embedding layer which transforms the $10 \times 10 \times 1$ integer input array into a continuous representation of size $10 \times 10 \times 32$, a convolutional layer with 32 filters of size $3 \times 3$, and another $3 \times 3$ convolutional layer with 64 filters.

- The second branch takes as input a local map with size $5 \times 5$ centered around the agent's position. The map encoding is the same as for the first branch and an embedding layer is followed by convolutional layers with the same structure as the previous ones.

- The third branch is structured like the second, but with a local map of size $3 \times 3$.

- The final branch takes as input an array of 11 discrete values containing information about the agent and the player:

    - agent HP in the range [0,20];
    - the potion currently in the agent's inventory;

- the melee weapon currently in the agent's inventory;
- ranged weapon in the agent's inventory;
- a value indicating whether the agent has an active buff;
- a value indicating whether the agent can perform a ranged attack and in which direction;
- player HP in the range [0,20];
- the potion currently in the player's inventory;
- the melee weapon in the player's inventory;
- the ranged weapon in the player's inventory; and
- a value indicating whether the player has an active buff.

This layer is followed by an embedding layer of size 64 and a fully-connected (FC) layer of size 256.

The outputs of all branches are concatenated to form a single vector which is passed through an FC layer of size 256; we add a *one-hot* representation of the action taken at the previous step, and the resulting vector is passed through an LSTM layer.

The final output of the net is a probability distribution over the action space (like all policy-based methods such as PPO). Instead of taking the action with the highest probability, we sample the output, thus randomly taking *one of the most probable actions*. This behavior lets the agent make some mistakes during its interaction with the environment, guaranteeing *imperfection* and avoids the agent getting stuck in repetitive loops of the same moves.

With this model we also propose two novel solutions that have improved the quality of the agent behavior, overcoming some of the challenges of DRL in real applications:

- **Global vs local view**: we discovered that the use of both global and local map representations improves the score achieved by the agent and the overall quality of its behavior. The combination of the two representations helps the agent evaluate both the general situation of the environment and the local details close to it; we use only two levels of local maps, but for a more complex situation game developers could potentially use more views at different scales;

- **Embeddings**: the embedding layers make it possible for the network to learn continuous vector representations for the meaning of and differences between integer inputs. Of particular note is the embedding of the last branch of the network, whose inputs have their own ranges distinct from each other, which helps the agent distinguish the contents of two equal but semantically different integer values. For instance:

    - agent HP $\in [0, 20]$;

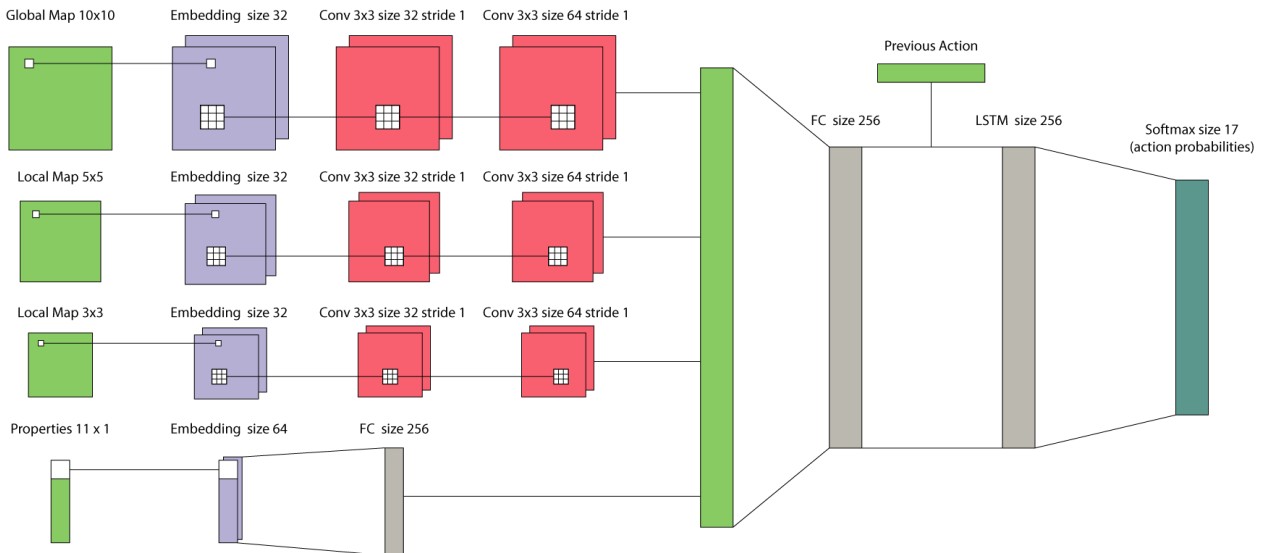

*Figure 2.* The policy network used for NPCs in DeepCrawl. The net has four input branches: the first takes as input the global $10 \times 10$ map, the second and third take as input a local map centered around the agent with different sizes, and the fourth takes as input an array of agent and player properties. All branches contain an embedding layer that transforms the discrete inputs into continuous ones. The first three branches consist of two convolutional layers, the fourth of a fully-connected layer, and the outputs of all branches are then concatenated together before a fully-connected layer and an LSTM layer, which also receives a one-hot representation of the previous action taken by the agent (see section 4.1 for more details.)

- potion $\in [21, 23]$;
- melee weapon $\in [24, 26]$;
- etc.

## 4.2. Reward shaping

When shaping the reward function for training policy networks, to satisfy the *model-free* requirement we used an extremely sparse function:

$$R(t) = -0.01 + \begin{cases} -0.1 & \text{for an impossible move} \\ +10.0 * \text{HP} & \text{for the win} \end{cases},$$

$$\tag{1}$$

where HP refers to the normalized agent HPs remaining at the moment of defeating an opponent. This factor helps the system to learn as fast as possible the importance of HP: winning with as many HP as possible is the implicit goal of Roguelikes.

## 4.3. Network and training complexity

All training was done on an NVIDIA 1050ti GPU with 4GB of RAM. On this modest GPU configuration, complete training of one agent takes about two days. However, the reduced size of our policy networks (only about 5.5M parameters in the policy and baseline networks combined) allowed us to train multiple agents in parallel. Finally, the trained system

needs about 12MB to be stored. We remind though that more agents of the same type can use the same model: therefore this system does not scale with the number of enemies, but only with the number of different classes.

These are the main elements of the DeepCrawl NPC model, many of which are directly related to the desiderata outlined in section 3. We now turn to our implementation and the technologies that make DeepCrawl possible.

## 5. Implementation

In this chapter we describe how the DeepCrawl policy networks were trained as well as the technologies used to build both the DRL system and the game.

### 5.1. Tensorforce

Tensorforce (Kuhnle et al., 2017; Schaarschmidt et al., 2018) is an open-source DRL framework built on top of Google's TensorFlow framework, with an emphasis on modular, flexible library design and straightforward usability for applications in research and practice.

Tensorforce is agnostic to the application context or simulation, but offers an expressive state- and action-space specification API. In particular, it supports and facilitates working with multiple state components, like our global/local map plus property vector, via a generic network configuration

| | agent HPs | enemy HP | loot quantity |
|---|---|---|---|
| Phase 1 | 20 | 1 | 20% |
| Phase 2 | [5, 20] | 10 | 20% |
| Phase 3 | [5, 20] | [10, 20] | 20% |
| Phase 4 | [5, 20] | [10, 20] | [10%, 20%] |
| Phase 5 | [5, 20] | [10, 20] | [5%, 20%] |

*Figure 3.* Curriculum used for training all agents. Left: a training timeline showing how long each curriculum phase lasts as a percentage of total training steps. Right: the changing generation parameters of all the curriculum phases. The numbers in parentheses refer to a random number in that; the loot quantity is a percentage of the empty tiles in the room (e.g. 20% loot quantity indicates a 20% chance of generating loot on each empty tile). The NPC intrinsic properties (ATK and DEF) stay fixed for the entire duration of the training. These parameters can be modified by developers to differentiate distinct behavioral classes, as we explain in section 5.4.

interface which is not restricted to simple sequential architectures only. Moreover, the fact that Tensorforce implements the entire RL logic, including control flow, in portable TensorFlow computation graphs makes it possible to export and deploy the model in other programming languages, like C# as described in the next section.

## 5.2. Unity and Unity ML-Agents

The DeepCrawl prototype was developed with Unity (Unity, 2019), a cross-platform game engine for the creation of 2D/3D multimedia contents. The Unity Machine Learning Agents Toolkit (Unity ML-Agents) (Juliani et al., 2018) is an open source plugin that enables games and simulations to serve as environments for training intelligent agents. This framework allows external Python libraries to interact with the game code and provides the ability to use pre-trained graph directly within the game build thanks to the TensorFlowSharp plugin (Icaza, 2019).

## 5.3. Training setup

To create agents able to manage all possible situations that can occur when playing against a human player, a certain degress of randomness is required in the procedurally-generated environments: the shape and the orientation of the map, as well as the number of impassable and collectible objects and their positions are random; the initial position of the player and the agent is random; and the initial equipment of both the agent and the player is random.

In preliminary experiments we noticed that agents learned very slowly, and so to aid the training and overcome the problem of the sparse reward function, we use curriculum learning (Bengio et al., 2009) with phases shown in figure 3. This technique lets the agent gradually learn the best moves to obtain victory: for instance, in the first phase it is very easy to win the game, as the enemy has only 1 HP and only one attack is needed to defeat it; in this way the model

can learn to reach its objective without worrying too much about other variables. As training proceeds, the environment becomes more and more difficult to solve, and the "greedy" strategy will no longer suffice: the agent HP will vary within a range of values, and the enemy will be more difficult to defeat, so it must learn how to use the collectible items correctly and which attack is the best for every situation. In the final phases loot can be difficult to find and the HP, of both agent enemy, can be within a large range of values: the system must develop a high level of strategy to reach the end of the game with the highest score possible.

The behavior of the enemies agents are pitted against is of great importance. To satisfy requirements defined in section 3.2, the enemy always makes random moves during training of agent NPCs. In this way, the agent sees all the possible actions that a user might perform, and at the same time it can be trained against a limited enemy with respect of human capabilities. This makes the agent beatable in the long run, but still capable of offering a tough challenge to the human player.

## 5.4. Training results

The intrinsic characteristic values for NPCs must be chosen before training. These parameters are not observed by the system, but offer an easy method to create different types of agents. Changing the agent's ATK, DEF or DEX obliges that agent to extrapolate the best strategy based on its own characteristics. For DeepCrawl we trained three different combinations:

- **Archer**: ATK = 0, DEX = 4 and DEF = 3;
- **Warrior**: ATK = 4, DEX = 0 and DEF = 3; and
- **Ranger**: ATK = 3, DEX = 3 and DEF = 3.

For simplicity, the opponent has always the same characteristics: ATK = 3, DEX = 3 and DEF = 3.

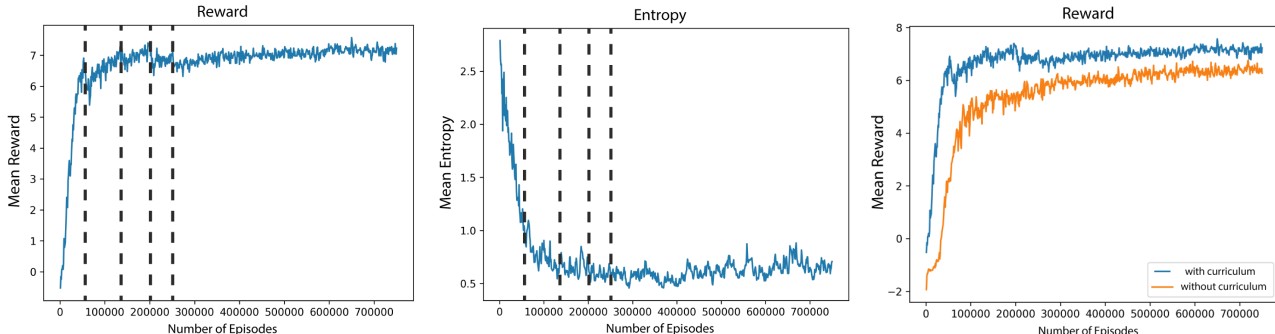

*Figure 4.* Plots showing metrics during the training phase for the warrior class as a function of the number of episodes. From left to right: the evolution of the mean reward, the evolution of the entropy, and the difference between the training with and without curriculum. The dashed vertical lines on the plots delineate the different curriculum phases.

To evaluate training progress and quality, we performed some quantitative analysis of the evolution of agent policies. In figure 4 we show the progression of the mean reward and entropy for the warrior class as a function of the number of training episodes. The other two types of agents follow the same trend. In the same figure we show the difference between training with and without curriculum learning. Without curriculum, the agent learns much slower compared to multi-phase curriculum training. With a curriculum the agent achieves a significantly higher average reward at the end of training.

### 5.5. PPO and hyperparameters

To optimize the policy networks we used the PPO algorithm (Schulman et al., 2017). One agent rollout is made of 10 episodes, each of which lasts at most 100 steps, and it may end either achieving success (i.e. agent victory), a failure (i.e. agent death) or reaching the maximum steps limit. At the end of 10 episodes, the system updates its weights with the episodes just experiences. PPO is an Actor-Critic algorithm with two functions that must be learned: the policy and the baseline. The latter has the goal of a normal state value function and, in this case, has the exactly same structure as the policy network show in figure 4.1.

Most of the remaining hyper-parameters values were chosen after many preliminary experiments made with different configurations: the policy learning rate $lr_p = 10^{-6}$, the baseline learning rate $lr_b = 10^{-4}$, the agent exploration rate $\epsilon = 0.2$, and the discount factor $\gamma = 0.99$.

## 6. Playability evaluation

To evaluate the DeepCrawl prototype with respect to our desiderata, we conducted playability test as a form of qualitative analysis. The tests were administered to 10 candidates, all passionate videogamers with knowledge of the domain; each played DeepCrawl for sessions lasting about

60 minutes. Then, each player was asked to answer a Single Ease Question (SEQ) questionnaire. All the questions were designed to understand if the requirements laid out in section 3.2 had been met and to evaluate the general quality of DeepCrawl. Table 1 summarizes the results.

We reconsider here each of the main requirements we discussed above in section 3.2 in light of the player responses:

- **Credibility**: as shown by questions 3, 4, and 5, the agents defined with this model offer a tough challenge to players; the testers perceived the enemies as intelligent agents that follow a specific strategy based on their properties.

- **Imperfection**: at the same time, questions 1 and 2 demonstrate that players are confident they can finish the game with the proper attention and time. So, the agents we have trained seem far from being superhuman, but they rather offer the right amount of challenge and result in a fun gameplay experience (question 14).

- **Model Free**: questions 5 and 12 show that, even with a highly sparse reward, the model is able to learn a strategy without requiring developers to define specific behaviors. Moreover, question 13 indicates that the agents implemented using DRL are comparable to others in other Roguelikes, if not better.

- **Variety**: the testers stated that the differences between the behaviors of the distinct types of agents were very evident, as shown by question 6. This gameplay element was much appreciated as it increased the level of variety and fun of the game, and improved the general quality of DeepCrawl.

## 7. Conclusions and future work

In this we presented a new DRL framework for development of NPC agents in video games. To demonstrate the potential of DRL in video game production, we designed and

*Table 1.* Results of the SEQ questionnaire administered after the playability tests. Players answered each question with a value between 1 (strongly disagree) and 7 (strongly agree).

| N° | QUESTION | MEAN | σ |
|----|----------|------|---|
| 1 | WOULD YOU FEEL ABLE TO GET TO THE END OF THE GAME? | 5.54 | 1.03 |
| 2 | AS THE LEVEL INCREASES, HAVE THE ENEMIES SEEMED TOO STRONG? | 4.63 | 0.67 |
| 3 | DO YOU THINK THAT THE ENEMIES ARE SMART? | 5.72 | 0.78 |
| 4 | DO YOU THINK THAT THE ENEMIES FOLLOW A STRATEGY? | 6.18 | 0.40 |
| 5 | DO YOU THINK THAT THE ENEMIES DO COUNTERINTUITIVE MOVES? | 2.00 | 0.63 |
| 6 | DO THE DIFFERENT CLASSES OF ENEMIES HAVE THE SAME BEHAVIOR? | 1.27 | 0.46 |
| 7 | ARE THE MEANING OF THE ICONS AND WRITING UNDERSTANDABLE? | 5.72 | 1.67 |
| 8 | ARE THE INFORMATION GIVEN BY THE USER INTERFACE CLEAR AND ENOUGH? | 5.54 | 1.21 |
| 9 | ARE THE LEVEL TOO BIG AND CHAOTIC? | 2.00 | 1.34 |
| 10 | ARE THE OBJECTS IN THE MAP CLEARLY VISIBLE? | 5.81 | 1.66 |
| 11 | DO YOU THINK THAT IS USEFUL TO READ THE ENEMY'S CHARACTERISTICS? | 6.90 | 0.30 |
| 12 | HOW MUCH IS IMPORTANT TO HAVE A GOOD STRATEGY? | 6.90 | 0.30 |
| 13 | GIVE A GENERAL VALUE TO ENEMY ABILITIES COMPARED TO OTHER ROGUELIKE GAMES | 6.00 | 0.77 |
| 14 | IS THE GAME ENJOYABLE AND FUN? | 5.80 | 0.87 |
| 15 | DOES THE APPLICATION HAVE BUGS? | 1.09 | 0.30 |

implemented a new Roguelike called DeepCrawl that uses the model defined in this article with excellent results. The current versions of the agents work very well, and the model supports numerous agents types only by changing a few parameters before starting training. We feel that DRL brings many advantages to the table commonly used techniques like finite state machines or behavior trees.

Agents in DeepCrawl do not use predefined strategies, but are able to extrapolate them autonomously. This makes them more intelligent and unpredictable with respect to classical techniques. Moreover, this model is not necessarily limited to DeepCrawl, but can be potentially used in any Roguelike sharing the same general gameplay mechanics of this prototype.

Despite the positive reception of DeepCrawl by playtesters, there remain many open problems in applying DRL to video game design. One of these is the general scalability of the system: the neural network described here works reasonably well in a limited and well-defined context such as DeepCrawl, but in a complex development process it can be difficult to manage the many changes made on-the-fly during production. To address this problem and to improve the scalability and the general efficiency of the system our efforts are leading in several directions:

- **Hierarchical Control**: agent strategies are often composed of one or more task levels (e.g. inventory manipulation, movement, combat, etc.). This suggests the use of hierarchical control, where separate policy networks are dedicated to each sub-task. It may also be possible to learn this hierarchical structure, as in the FuN architecture (Sasha Vezhnevets et al., 2017);

- **Fine-tuning**: to deal with gameplay changes during the development and design process, we are exploring

fine-tuning in DRL, where the behavior of a generic, pre-trained agent is specialized by replacing a few layers of its policy network and restarting training;

- **Meta-learning**: AI for NPCs in video games is an excellent example of models that must be capable of adapting quickly to new situations and states which differ significantly from those seen during training. Recent works in meta-learning have shown that fast adaptation in DRL is possible (Finn et al., 2017; Wang et al., 2016), and we believe that this ability can improve the scalability of the DRL system and will be needed to perform well in complex scenarios; and

- **Learning from Human Preference**: it is essential to allow designers to have some control over the behavior of the agents. For this, we are looking at applying a preference-based method (Christiano et al., 2017) that allows agents to learn from games based on a combination of human preference learning, in order to provide designers with a tool that allows to easily specify desired and/or undesired NPC behaviors.

Videogaming is a mainstream and transversal form of entertainment these days. The recent and highly-publicized successes of DRL in mimicking or even surpassing human-level play in games like Go and DOTA have not net been translated into effective tools for use in developing game AI. The DeepCrawl prototype is a step in this direction and shows that DRL can be used to develop credible – yet imperfect – agents that are model-free and offer variety to gameplay in turn-based strategy games like Roguelikes. We feel that DRL, with some more work towards rendering training scalable and flexible, can offer great benefits over classical, hand-crafted agent design that dominates the industry.

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
