# OpenReview forum: "DeepCrawl: Deep Reinforcement Learning for Turn-based Strategy Games"
_ICML.cc/2019/Workshop/RL4RealLife — Submitted to RL4RealLife 2019_

### Official Review · AnonReviewer2 · 2019-05-23
**Comments**

**Rating:** 3
**Confidence:** 4

**Review:**

This paper deals with the problem of developing behavioral models for NPCs in video games. This is an interesting topic. Here the major difference with other RL applications is to make NPCs relatively competitive. My biggest concern for this paper is whether the proposed method has any advantage, in terms of playability and scalability, comparing to classical technique.

Pros:
1. This paper is easy to read and clearly motivated. The prototype is well designed and fully playable.
2. The user evaluation looks promising. It seems that the proposed method could generate different policies for various types of NPCs. Potentially, this could be a valuable property when designing games with a large number of NPCs.

Cons:
1. Using RL to train behavioral models for NPCs is not a new topic. This paper should cite relevant work including:

Christopher Amato and Guy Shani. High-level reinforcement learning in strategy games. In Proceedings of the 9th International Conference on Autonomous Agents and Multiagent Systems: volume 1 - Volume 1 (AAMAS '10), Vol. 1. International Foundation for Autonomous Agents and Multiagent Systems, Richland, SC, 75-82.

2. The authors argue that it is difficult to scale traditional AI solutions to behavior modeling. However, it is well-known that model-free reinforcement learning also suffers from large sample space. It is unclear how the authors plan to address this problem in larger games.

3. Also, it is necessary to conduct playability evaluations with respect to NPCs modelled by traditional solutions (e.g. behavior trees, finite state machines) as baselines. It would give us some idea about the relative quality of NPCs trained by the proposed algorithm.

---

### Official Review · AnonReviewer1 · 2019-05-24
**Good prototype but is it related to the real life?**

**Rating:** 2
**Confidence:** 5

**Review:**

This paper introduces DeepCrawl, a prototype supporting mobile OSs. The authors use a policy network in deep reinforcement learning to train their agents. The reviewer agrees that the video game industry has a great potential to grow up using AI technologies. However, there are a few concerns as below.
1.	Scope: The reviewer is not convinced how this paper is fit in the scope of RL for Real Life Workshop. This is the traditional video game-based simulation, and there is no impact on real life from the trained agent.
2.	 Novelty: The reviewer believes that there is no novelty in terms of the reinforcement learning algorithm. Rather, this paper has a strong contribution to building a new prototype which can be a future benchmark. However, the manuscript could not convince readers such contribution in a current form.
3.	Writing: The manuscript is not well-organized. It is hard to follow. It might be better to have a neat storyline in the paper.

---

### Decision · Program_Chairs · 2019-05-28

Reject